# The Morphology, Genetic Diversity, and Distribution of *Ulva meridionalis* (Ulvaceae, Chlorophyta) in Chinese Seas

Meijuan Hu [1,†], Shuang Zhao [1,†], Jinlin Liu [1,†], Yichao Tong [1], Zhangyi Xia [1], Jing Xia [1,2], Shuang Li [1], Yuqing Sun [1], Jiaxing Cao [1] and Jianheng Zhang [1,3,4,*]

1   College of Marine Ecology and Environment, Shanghai Ocean University, Shanghai 201306, China
2   School of Oceanography, Shanghai Jiao Tong University, Shanghai 200030, China
3   Key Laboratory of Exploration and Utilization of Aquatic Genetic Resources, Ministry of Education, Shanghai Ocean University, Shanghai 201306, China
4   Co-Innovation Center of Jiangsu Marine Bio-industry Technology, Jiangsu Ocean University, Lianyungang 222005, China
*   Correspondence: jh-zhang@shou.edu.cn
†   These authors contributed equally to this work.

**Abstract:** Green tides originate from the rapid growth of green macroalgae and their large accumulation. In the past few decades, the severity and frequency of green tides have increased and the range of their geographical distribution has widened. In recent years, *Ulva meridionalis* Horimoto et Shimada has been reported in many countries. This species has stable morphological characteristics, and its length can reach 3 m in indoor cultures. Its cells contain pyrenoids, and the sporangium and gametangium of each cell contain 8 spores and 16 gametes, respectively, which confer a high proliferation potential. The phylogenetic tree constructed in this study showed that the Internal Transcribed Spacer sequence identified *U. meridionalis* with a high identification reliability, and the genetic relationship between *U. meridionalis* and *Ulva pertusa* in the ITS sequence was close. The haplotype network analysis clarified the relationship of the *U. meridionalis* samples collected from four different sea areas in China and indicated that they were closely related. Five haplotypes were identified: Hap_2 and Hap_1 were the most frequent, and they were also the haplotypes shared among the three groups. The degree of subspecies formation was not reached among these *U. meridionalis* samples collected from the Chinese seas. Up to 20 years ago, *U. meridionalis* had only been recorded in Japan. After 2011, it has been found to be widely distributed in the United States, China, French New Caledonia, French Polynesia, and Australia, where it proliferates. It has spread as a new kind of green tide-forming macroalga. The present study found that *U. meridionalis* is widely distributed in the Chinese seas; specifically, there have been small-scale blooms in the Bohai Sea, the Yellow Sea, and the South China Sea. Further investigations should focus on establishing whether *U. meridionalis* will cause large-scale green tide events in the future.

**Keywords:** *Ulva meridionalis*; Ulvaceae; algal bloom; Yellow Sea; intertidal zone

## 1. Introduction

A green tide is an ecological phenomenon caused by the rapid proliferation and aggregation of green macroalgae after they detach from specific substrata and reach a free-floating state. Most green tides occur in semi-closed sea areas, such as estuaries or inner bays. When a green tide occurs, large amounts of green macroalgae float on the sea. Green macroalgae consume a large number of dissolved oxygen and release toxins when decaying, which affect other marine organisms and change the structure and function of the marine ecosystem [1–3]. Eventually, the green macroalgae will land under the influence of wind and ocean currents, which will not only affect the tourism of coastal cities, but will also affect the aquaculture industry, causing huge economic losses. At the same time, the bad smell of green macroalgae may also be harmful to human health [4,5]. There is

a higher risk of biological invasion associated with green algae than with red and brown algae [6]. *Ulva* species are common to the global oceans and estuaries, and they are usually dominant in green tides due to their rapid growth, diverse reproduction modes, and strong adaptability to the environment [7,8].

*Ulva* spp. can spread to different marine areas through ballast water and seafood trade, or by attaching to ships and other floating objects [9–12], and they tend to bloom in new habitats as invasive species [13]. At present, about 100 *Ulva* species are known in the world [14–16], some of which are the main dominant species causing green tide outbreaks [17,18]. At the same time, as marine eutrophication continues to intensify worldwide, green tides are becoming more frequent at a global scale [19]; for example, South Africa [20], the United States [21], Japan [22], the Philippines [23], Ireland [24], and France [25] have seen outbreaks of green tides. In addition, *Ulva* can cause green tides not only in marine ecosystems, but also in freshwater ecosystems. such as *U. pilifera* [26] and *U. flexuosa* subsp. paradoxa [27].

Approximately 20 *Ulva* species have been recorded in China [28], and among those responsible for the outbreaks are *Ulva prolifera* O.F. Müller, *Ulva compressa* Linnaeus, *Ulva intestinalis* Linnaeus, *Ulva linza*, *Ulva flexuosa* Wulfen, and *Ulva australis* Areschoug (as *U. pertusa* Kjellman) [29–33]. However, over the last 10 years, a new species causing green tides has emerged: *Ulva meridionalis* Horimoto et Shimada. *U. meridionalis* is native to the tropical and subtropical areas of the Indian and Pacific Oceans [34,35], but can also grow in temperate areas and is commonly found on the gravel or reefs of intertidal zones, bays, and estuaries. In 2000, samples of *U. meridionalis* were collected for the first time in the estuary of the Todoroki River, Japan, and the first small-scale outbreak occurred in its estuary area [36]. In 2015, a large amount of *U. meridionalis* was found in the intertidal zone of Townsville, Queensland, Australia, accounting for 94% of the *Ulva* specimens sampled [37]. In 2020, a large biomass of green macroalgae with high species diversity was found in the intertidal zone of locations along the eastern coast of the United States and the Gulf of Mexico. This study reported the existence of *U. meridionalis* in North America for the first time, but the species did not cause green tides in the region [12]. In 2022, for the first time, researchers detected a considerable biomass of *U. meridionalis* in Oceania. This species might have emerged in this region due to natural diffusion or maritime traffic [35]. At present, there is an increasing trend of green tides at the regional scale in Oceania.

Over the past five years, *U. meridionalis* has also been found in China [38,39]. At present, the species has mainly bloomed in the Bohai Sea (Figure 1A), the Yellow Sea (Figure 1B), and the South China Sea (Figure 1C), but no outbreaks have been recorded in the East China Sea. Current observations show that *U. meridionalis* mainly grows near intertidal mudflats, dikes, offshore plastic waste, and sewage outfalls, where it becomes the dominant species. In November 2018, a *U. meridionalis* outbreak occurred in the South China Sea [39], and this species accounted for about 92% of the biomass of algal blooms. Similarly, in 2021, a green tide consisting primarily of *U. meridionalis* occurred for the first time in Yingkou, Dalian, Liaoning Province. In addition, *U. meridionalis* drops from the attached state and floats in the local sea area (e.g., the Bohai Sea area), and a large quantity of *U. meridionalis* has been found in offshore mariculture ponds [40] The aim of the present study was to investigate *U. meridionalis* using basic biology, molecular identification, and phylogenetic methods for analysis, and to introduce the basic biological characteristics of *U. meridionalis*, the species relationship of *U. meridionalis* in the four sea areas of China, and the current distribution of *U. meridionalis* in the world. In addition, this study may contribute to further understanding the regularity of green tides caused by *U. meridionalis* and providing a dataset and reference for the control of future outbreaks.

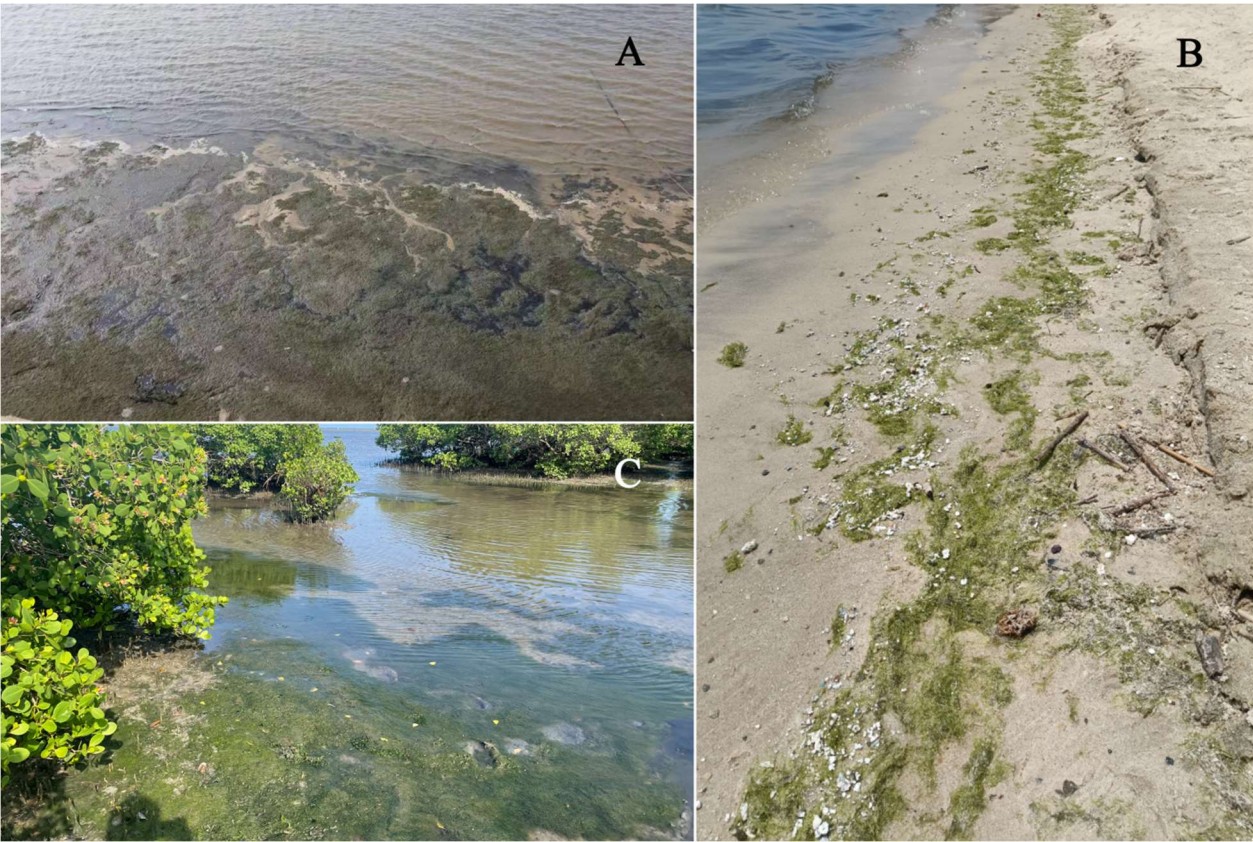

**Figure 1.** Green tide outbreaks dominated by *U. meridionalis* in (**A**) Yingkou, Dalian City, Liaoning Province, August 2021; (**B**) Qidong, Nantong City, Jiangsu Province, July 2021; (**C**) Wenchang City, Hainan Province, May 2021.

## 2. Materials and Methods

### 2.1. Sample Collection and Molecular Identification

From May to November 2021, green macroalgae were collected in the intertidal zones of the Bohai Sea, the Yellow Sea, the East China Sea, and the South China Sea. The samples were washed with sterilized seawater, and all epiphytes on the surface of algae were removed with a sterilized brush. Then, the samples were numbered for subsequent molecular identification.

An Ezup Column Plant Genomic DNA Purification Kit was used to extract DNA from the samples. The ITS primer sequences were ITS-F (5'-TCTTTGAAACCGTATCGTGA-3') and ITS-R (5'-GCTTATTGATATGCTTAAGTTCA GCGGGT-3') [41]. The ITS region was amplified via polymerase chain reaction (PCR). The PCR reaction mix contained 2 μL of sample DNA, 2 μL of upstream primer, 2 μL of downstream primer, 25 μL of PCR-Mix, and 19 μL of dd-$H_2O$. The PCR amplification procedure was performed under the following conditions: initial denaturation at 94 °C for 5 min, 30 cycles of denaturation at 94 °C for 1 min, primer annealing at 60 °C for 1 min, extension at 60 °C for 2 min, and holding at 60 °C for 10 min. The samples were stored at 4 °C. All the high-quality PCR products were sent to Sangong Biotechnology Co. Ltd. (Shanghai, China) for DNA sequencing, and all the samples identified as *U. meridionalis* were used for subsequent research (Table 1).

### 2.2. Morphological and Microscopic Observations

Through observation, it was found that the morphological characteristics of *U. meridionalis* in the four sampled areas were consistent, and the samples collected from Liaoning Province in the Bohai Sea area were selected (Station number: CN9) (Table 1) to conduct further basic biological observations. An optical microscope (E200, Nikon, Tokyo, Japan) was used to observe the thallus structure of the algae in detail, and the length and



width of cells at the basal, middle, and upper regions were measured. The pyrenoids and morphology of algal cells were photographed. After using the punching method to induce algal release [42] and then the release of tetraflagellate meiospore, biflagellate gametes were observed and photographed under an optical microscope. Then, single-cell culture of *U. meridionalis* was performed at 20 °C, 62.5–75 μmol m$^{-2}$ s$^{-1}$, and a 12 h light: 12 h dark photoperiod (12L:12D) for 65 days; the culture medium was changed every 3 days. Photographs of *U. meridionalis* were taken with a Sony camera (7M3, SONY, Tokyo, Japan) as morphological records. The preparation method of the herbarium documentation was according to Gao & Liu [43], and we digitized it according to the document description of Quick Response code usage. Voucher specimen was deposited in the herbarium of the College of Marine Ecology and Environment, Shanghai Ocean University (Figure 2A).

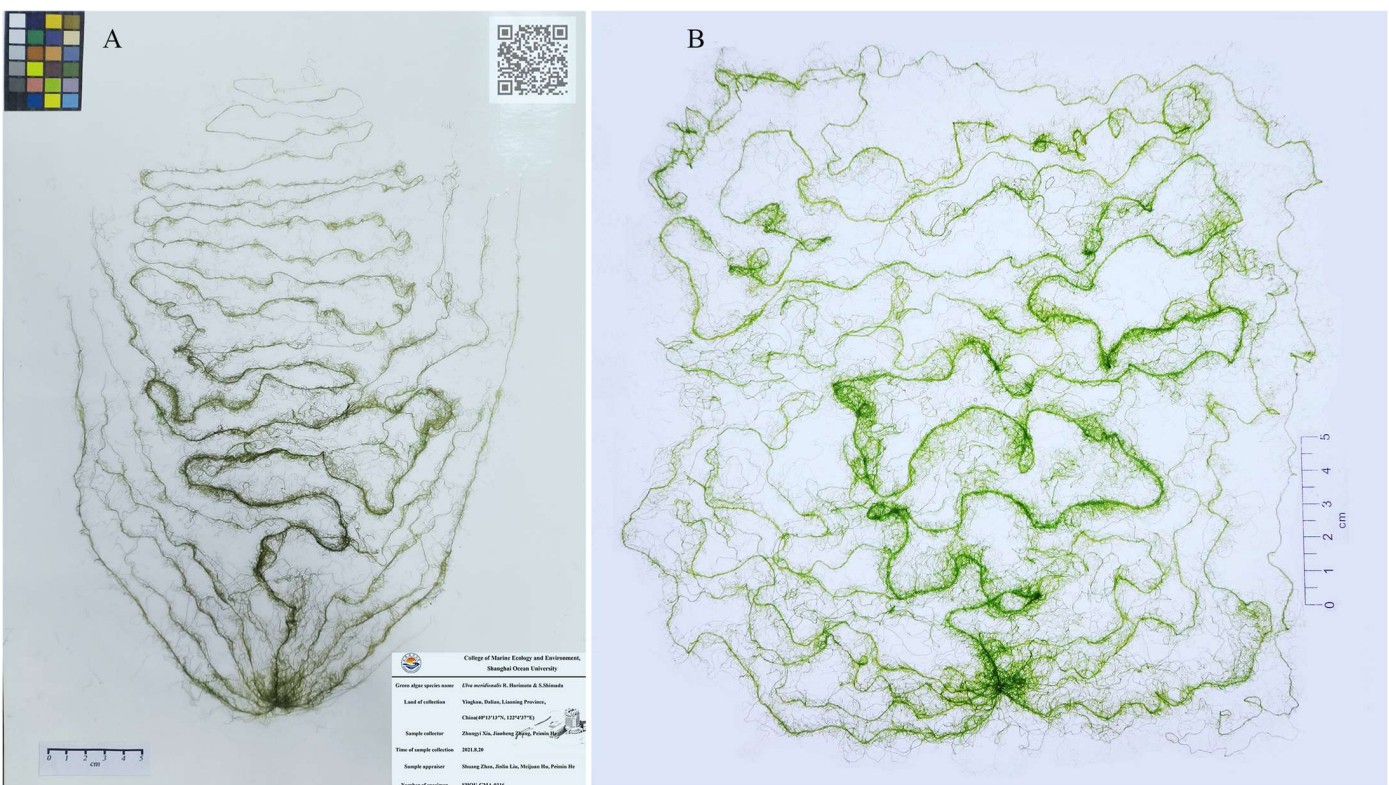

**Figure 2.** Morphological photography and herbarium documentation: (**A**) specimen of *Ulva meridionalis* (Voucher code: SHOU-GMA-0316), and (**B**) morphology of the thallus under indoor culture conditions.

### 2.3. Morphological and Microscopic Observations

The *U. meridionalis* ITS sequences were obtained for the following samples collected from the four sea areas of this study: (1) the Bohai Sea (Sample number: CN9-1-China; CN9-2-China; CN9-3-China); (2) the Yellow Sea (Sample number: CN1-1-China; CN2-1-China; CN2-2-China; CN3-1-China; CN3-2-China; CN4-1-China; CN4-2-China; CN5-1-China; CN6-1-China; CN6-2-China; CN7-1-China; CN7-2-China; CN7-3-China; CN7-4-China); (3) the East China Sea (Sample number: CN8-1-China); and (4) the South China Sea (Sample number: CN10-1-China; CN10-2-China; CN10-3-China) (Table 1).

The following sequences, which were similar to that of *U. meridionalis*, were downloaded from the National Biotechnology Center (National Center for Biotechnology Information, NCBI) database: MK426969 *U. meridionalis* (Zhanjiang, Guangdong, South China Sea), MK426968 *U. meridionalis* (Zhanjiang, Guangdong, South China Sea), MK426970 *U. meridionalis* (Yancheng, Jiangsu, Yellow Sea), MK426972 *U. meridionalis* (Rizhao, Shandong, Yellow Sea), MK426971 *U. meridionalis* (Yancheng, Jiangsu, Yellow Sea), and other

related species (HQ902008 *U. pertusa*, MG017465 *U. prolifera*, AJ000203 *U. linza*, EU933981 *U. compressa*, HM031176 *U. flexuosa*, KC661326 *U. erecta* (Lyngbye) Fries, KC661346 *U. fasciata* Delile, MF139299 *U. intestinalis*, MZ596117 *U. tepida* Y. Masakiyo et S. Shimada). Then, a phylogenetic tree was constructed in MEGA7.0.14 using the Maximum Likelihood (ML) method. In addition, DNASP5 software [44,45] was used to count and analyze the haplotypes of the 26 samples (21 samples from the four sea areas in this study and five samples from the NCBI), and Network 5.0 was used to perform the haplotype network analysis and examine the evolutionary relationship among the haplotypes in each sample.

**Table 1.** Geographical location of the *U. meridionalis* sample stations.

| Sample Numbers | Continent | Country and Region | Longitude | Latitude | Habitats | Reference |
|---|---|---|---|---|---|---|
| CN1 | Asia | Yancheng, Jiangsu Province, China | 120°01′06.372″ E | 34°25′14.940″ N | Attached in the intertidal zone | This study |
| CN2 | Asia | Yancheng, Jiangsu Province, China | 120°29′12.498″ E | 33°49′29.484″ N | Attached in the pond | This study |
| CN3 | Asia | Yancheng, Jiangsu Province, China | 120°31′26.526″ E | 33°46′42.630″ N | Floating in the pond | This study |
| CN4 | Asia | Nantong, Jiangsu Province, China | 121°17′22.158″ E | 32°27′37.794″ N | Attached to a seawall | This study |
| CN5 | Asia | Nantong, Jiangsu Province, China | 121°56′11.893″ E | 31°42′51.779″ N | Attached to the wastewater outlet | This study |
| CN6 | Asia | Nantong, Jiangsu Province, China | 121°39′14.670″ E | 32°04′28.422″ N | Attached in the intertidal mudflat | This study |
| S964 | Asia | Yancheng, Jiangsu Province, China | 120°45′ E | 33°15′ N | Attached in the pond | [46] |
| U246-8 | Asia | Yancheng, Jiangsu Province, China | 120°47′ E | 33°15′ N | Attached in the pond | [46] |
| S023 | Asia | Lianyungang, Jiangsu Province, China | 119°12′ E | 34°56′ N | Attached in the intertidal zone | [46] |
| CN8 | Asia | Ningbo, Zhejiang Province, China | 121°47′03.120″ E | 29°32′47.274″ N | Attached to a pier | This study |
| S199 | Asia | Ningbo, Zhejiang Province, China | 121°31′ E | 29°32′ N | Attached in the intertidal zone | [46] |
| N001 | Asia | Ningbo, Zhejiang Province, China | 121°33′ E | 29°26′ N | Attached in the intertidal zone | [46] |
| CN9 | Asia | Dalian, Liaoning Province, China | 122°05′01.198″ E | 40°14′54.776″ N | Floating in the intertidal zone | This study |
| CN10 | Asia | Wenchang, Hainan Province, China | 110°44′39.000″ E | 19°24′59.000″ N | Attached in the intertidal zone | This study |
| S654 | Asia | Haikou, Hainan Province, China | 110°32′ E | 20°01′ N | Attached in the intertidal zone | [46] |
| S203 | Asia | Haikou, Hainan Province, China | 110°16′ E | 20°01′ N | Attached in the intertidal zone | [46] |
| S981-1a | Asia | Qinhuangdao, Hebei Province, China | 119°37′ E | 39°55′ N | Attached in the intertidal zone | [46] |
| CN7 | Asia | Qingdao, Shandong Province, China | 120°06′37.756″ E | 36°12′26.316″ N | Floating in a pond | [38,42] |
| H940 | Asia | Qingdao, Shandong Province, China | 120°20′ E | 36°03′ N | Attached in the intertidal zone | [46] |
| S115 | Asia | Qingdao, Shandong Province, China | 120°21′ E | 36°02′ N | Attached in the intertidal zone | [46] |
| S027 | Asia | Qingdao, Shandong Province, China | 120°41′ E | 36°14′ N | Attached in the intertidal zone | [46] |
| S149 | Asia | Qingdao, Shandong Province, China | 120°10′ E | 35°53′ N | Attached in the intertidal zone | [46] |

**Table 1.** *Cont.*

| Sample Numbers | Continent | Country and Region | Longitude | Latitude | Habitats | Reference |
|---|---|---|---|---|---|---|
| S079 | Asia | Qingdao, Shandong Province, China | 120°20′ E | 36°03′ N | Attached in the intertidal zone | [46] |
| U261-3 | Asia | Rizhao, Shangdong Province, China | 119°36′ E | 35°29′ N | Attached in the intertidal zone | [46] |
| N177-1 | Asia | Weihai, Shangdong Province, China | 122°09′ E | 37°31′ N | Attached in the intertidal zone | [46] |
| N186-1 | Asia | Weihai, Shangdong Province, China | 122°11′ E | 37°30′ N | Attached in the intertidal zone | [46] |
| U225a | Asia | Zhanjiang, Guangdong Province, China | 110°04′ E | 20°17′ N | Attached in the intertidal zone | [46] |
| U277-1a | Asia | Zhanjiang, Guangdong Province, China | 110°25′ E | 21°13′ N | Attached to a seawall | [46] |
| S209 | Asia | Beihai, Guangxi Zhuang Autonomous Region, China | 109°09′ E | 21°24′ N | Attached in the intertidal zone | [46] |
| S217 | Asia | Shenzhen, Guangdong Province, China | 114°01′ E | 22°31′ N | Attached in the intertidal zone | [46] |
| S599 | Asia | Xiamen, Fujian Province, China | 118°05′ E | 24°34′ N | Attached in the pond | [46] |
| RH008 | Asia | Ishigaki Island, Okinawa Prefecture, Japan | 124°15′ E | 24°22′ N | Attached in the estuary | [36] |
| RH001-007 | Asia | Ishigaki Island, Okinawa Prefecture, Japan | 124°15′ E | 24°22′ N | Attached in the estuary | [36] |
| RH009-037 | Asia | Ishigaki Island, Okinawa Prefecture, Japan | 124°15′ E | 24°22′ N | Attached in the estuary | [36] |
| RH043-047 | Asia | Ishigaki Island, Okinawa Prefecture, Japan | 124°15′ E | 24°22′ N | Attached in the estuary | [36] |
| E16 | Asia | Tokushima, Tokushima Prefecture, Japan | 124°15′ E | 24°22′ N | Attached in the estuary | [36] |
| UNA00071829 | North America | Cedar Point, Alabama, America | 88°08′13.459″ W | 30°18′37.865″ N | Floating in the intertidal zone | [12] |
| UNA00071885 | North America | Coden, Alabama, America | 88°15′30.200″ W | 30°22′54.894″ N | Floating in the intertidal zone | [12] |
| UNA00072126 | North America | Choctawatchee Bay Bridge, Florida, America | 86°09′20.999″ W | 30°25′41.401″ N | Floating in the intertidal zone | [12] |
| UNA00072127 | North America | Choctawatchee Bay Bridge, Florida, America | 86°09′21.038″ W | 30°25′41.402″ N | Floating in the intertidal zone | [12] |
| UNA00072079 | North America | Copano Fishing Pier, Texa, America | 97°01′32.624″ W | 28°06′47.833″ N | Floating in the intertidal zone | [12] |
| UNA00072312 | North America | Charlotte Harbor, Florida, America | 82°04′17.893″ W | 26°57′22.950″ N | Floating in the intertidal zone | [12] |
| NOU218715 | Oceania | NA, Moindou, French New Caledonia | NA | NA | Attached in the intertidal zone | [35] |
| NOU218742 | Oceania | NA, Poe, French New Caledonia | NA | NA | Attached in the intertidal zone | [35] |
| NOU218743 | Oceania | NA, Poe, French New Caledonia | NA | NA | Attached in the intertidal zone | [35] |
| NOU218755 | Oceania | NA, Bourail, French New Caledonia | NA | NA | Attached in the intertidal zone | [35] |

**Table 1.** *Cont.*

| Sample Numbers | Continent | Country and Region | Longitude | Latitude | Habitats | Reference |
|---|---|---|---|---|---|---|
| NOU218803 | Oceania | NA, islet Double, French New Caledonia | NA | NA | Attached in the intertidal zone | [35] |
| NOU218847 | Oceania | NA, Poe, French New Caledonia | NA | NA | Attached in the intertidal zone | [35] |
| NOU218826 | Oceania | NA, Cap Goulevain, French Polynesia | NA | NA | Attached in the intertidal zone | [35] |
| NOU218832 | Oceania | NA, Cap Goulevain, French Polynesia | NA | NA | Attached in the intertidal zone | [35] |
| NOU218770 | Oceania | NA, Tahiti, French Polynesia | NA | NA | Attached in the intertidal zone | [35] |
| NOU215308 | Oceania | NA, Marquises, French Polynesia | NA | NA | Attached in the intertidal zone | [35] |
| NOU215309 | Oceania | NA, Marquises, French Polynesia | NA | NA | Attached in the intertidal zone | [35] |
| NOU218867 | Oceania | NA, Moorea, French Polynesia | NA | NA | Attached in the intertidal zone | [35] |
| TSV31 | Oceania | Townsville, Queensland, Australia | 146°48′39.999″ E | 19°14′42″ S | Attached in the intertidal zone | [37] |

## 3. Results

### 3.1. Basic Biological Characteristics of U. meridionalis

In a natural environment and under favorable growth conditions, the length and width of *U. meridionalis* can reach 45 cm and 0.8 cm, respectively. The alga is light green or yellow green, has a smooth surface, and is easy to break. Under indoor culture conditions, it can grow to a length of 3 m (Figure 2B). It has a tubular structure and an obvious main axis (Figure 3A), that folds in the wider part of the alga. In addition, with the growth of the algae, the middle of the tubular algae flattens and the upper and the lower layers of cells stick together. In the transverse section of the middle of the algae is a double-layer cell structure. However, the edge is still monolayered tubular structure. The transverse branches basically appear near the base, and transverse branch width gradually decreases toward the branch node direction (Figure 3B,C). For the collected samples, the length of algal cells in the base region was 9–30 μm, the width was 7–17 μm (Figure 3D), and the cells contained pyrenoids (Figure 3E). In the middle region, the length and width of algal cells was 8–23 μm and 6–16 μm, respectively (Figure 3F), and in the upper region, the same parameters measured 8–20 μm and 5–16 μm, respectively (Figure 3G). Both the length and width of *U. meridionalis* cells decreased from the base to the upper region. The chloroplasts of each cell contained 1–11 pyrenoids; specifically, the chloroplasts of the basal and the middle cells mainly had 4–5 and 4–6 pyrenoids, respectively, while there were 3–4 pyrenoids in the chloroplasts of the upper algal cells. The surface cells of *U. meridionalis* had a rounded triangle, rectangle, or polygon shape. Generally, after induced dispersal, *U. meridionalis* releases reproductive cells in about 3.5 days, which gradually disperse, and new sporophytes or gametophytes can form. The biflagellate gametes [which form in the gametangium (Figure 3H)] are oval, have a reddish eyespot (Figure 3I,J) and show positive phototaxis; tetraflagellate meiospores are ovate, have a reddish eyespot (Figure 3K), and show a negative phototaxis.

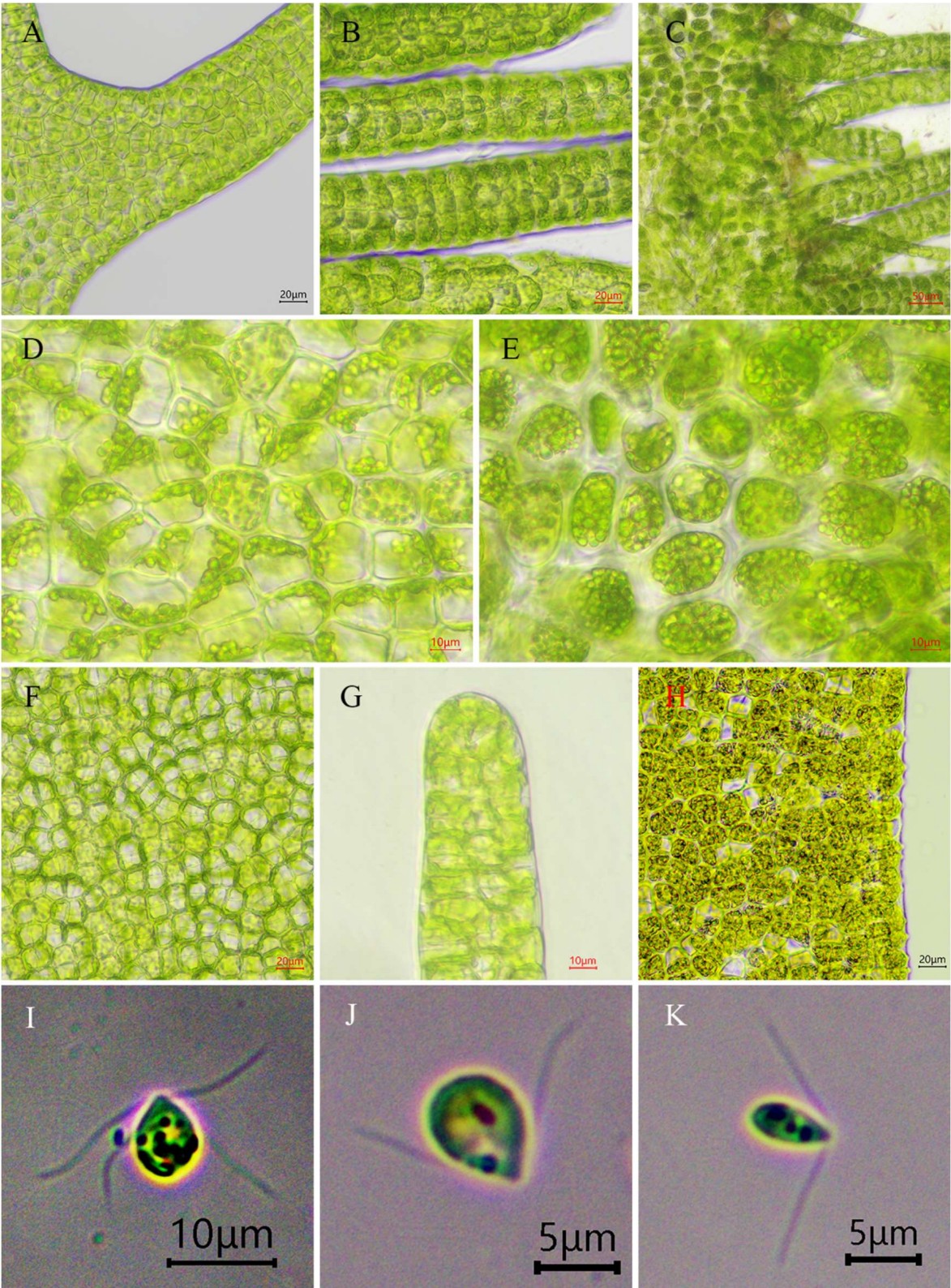

**Figure 3.** Microscopic observation of *Ulva meridionalis*: (**A**) main branch; (**B**,**C**) proximity of the horizontal branch to the base area; (**D**) surface view of cells in the basal portion; (**E**) cells containing many pyrenoids; (**F**) surface view of cells in the middle region; (**G**) surface view of cells in the upper region; (**H**) gametangium; (**I**) tetraflagellate meiospore; (**J**) biflagellate female gamete; and (**K**) biflagellate male gamete.

*3.2. Genetic Relationship among U. meridionalis Samples*

Based on the ITS-ML phylogenetic tree (Figure 4), all the samples examined in this study fell into seven evolutionary clusters, and HM031176 *U. flexuosa*, HQ902008 *U. pertusa*, and EU933981 *U. compressa* were grouped into separate evolutionary clusters.

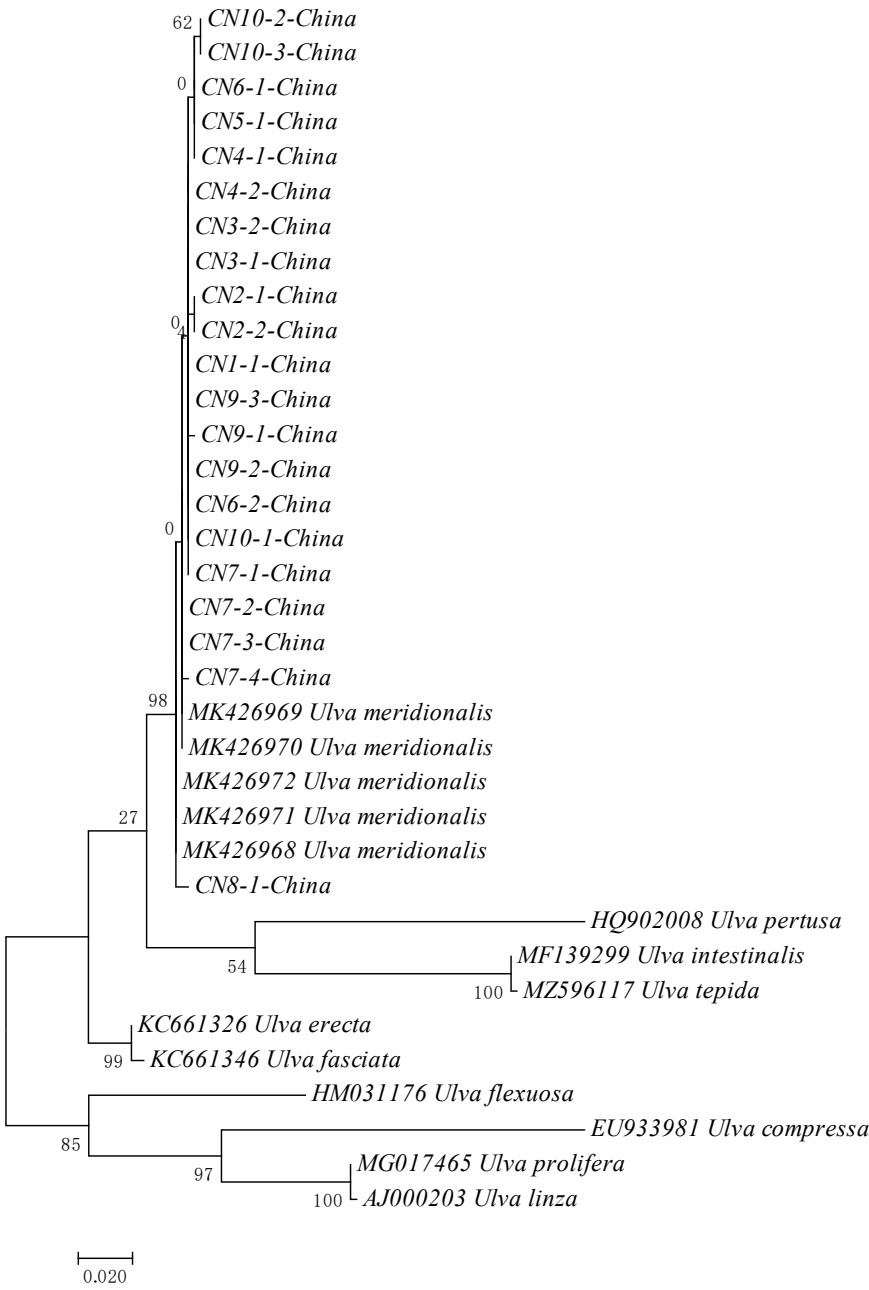

**Figure 4.** ML phylogenetic tree constructed based on the Kimura 2-parameter method.

The following sequences were grouped into single clusters: MF139299 *U. intestinalis* and MZ596117 *U. tepida*, (with a 100% sequence similarity); KC661326 *U. erecta* and KC661346 *U. fasciata* (sequence similarity, 99%); and MG017465 *U. prolifera* and AJ000203 *U. linza* (sequence similarity, 100%) (Figure 4). There are two reasons for this: On the one hand, although the ITS sequence as a barcode has a strong universality, it is not sufficient to distinguish few species under certain circumstances. On the other hand, it is also possible that the same species have different names.

All the *U. meridionalis* sequences were grouped into a single cluster (Sample numbers: CN9-1-China; CN9-2-China; CN9-3-China; CN1-1-China; CN2-1-China; CN2-2-China;

CN3-1-China; CN3-2-China; CN4-1-China; CN4-2-China; CN5-1-China; CN6-1-China; CN6-2-China; CN7-1-China; CN7-2-China; CN7-3-China; CN7-4-China; CN8-1-China; CN10-1-China; CN10-2-China; CN10-3-China; MK426969 *U. meridionalis*; and MK426970 *U. meridionalis*; MK426972 *U. meridionalis*; MK426971 *U. meridionalis*; MK426968 *U. meridionalis*), showing that the ITS bar code can identify *U. meridionalis* with a high accuracy. Among these sequences, the relationship between genetic distance and proximity was determined as follows: the *U. meridionalis* samples collected from Wenchang City, Hainan Province in the South China Sea (CN10-3-China; CN10-2-China) ⇌ the *U. meridionalis* samples collected from Nantong City, Jiangsu Province in the Yellow Sea (CN6-1-China) ⇌ the *U. meridionalis* samples collected from Yancheng City, Jiangsu Province in the Yellow Sea (CN5-1-China; CN4-1-China; CN4-2-China; CN3-1-China; CN3-2-China; CN2-2-China; CN2-1-China; CN1-1-China) ⇌ the *U. meridionalis* samples collected from in Dalian City, Liaoning Province in the Bohai Sea (CN9-2-China; CN9-1-China; CN9-3-China) ⇌ the *U. meridionalis* samples collected from Nantong City, Jiangsu Province in the Yellow Sea (CN6-2-China) ⇌ the *U. meridionalis* samples collected from Wenchang City, Hainan Province in the South China Sea (CN10-1-China) ⇌ the *U. meridionalis* samples collected from Qingdao City, Shandong Province in the Yellow Sea (CN7-1-China; CN7-2-China; CN7-3-China; CN7-4-China) ⇌ the *U. meridionalis* samples collected from Ningbo City, Zhejiang Province in the East China Sea (CN8-1-China). Individual, basic group differences were detected between the above *U. meridionalis* samples, but the degree of subspecies differentiation was not reached (Figure 4). Moreover, the ITS sequence indicated a close genetic relationship between *U. meridionalis* and *U. pertusa*, which is inconsistent with the results of the phylogeny of *U. meridionalis* based on organelle genome analysis.

The circle symbol represents one haplotype, and its size reflects the number of different haplotypes. The different colors represent different sources from which the haplotypes were derived. The network evolution diagram (Figure 5) shows that the *U. meridionalis* samples collected in the four sea areas in China are genetically close, and the haplotype evolution network shows a chain structure. A total of five haplotypes were detected: the samples from the East China Sea corresponded to the independent haplotype Hap_3; those from the South China Sea contained three haplotypes, Hap_1, Hap_2, and Hap_5; those from the Bohai sea presented two haplotypes, Hap_1 and Hap_2; and finally, the samples from the Yellow Sea contained three haplotypes, Hap_1, Hap_2, and Hap_4. Hap_2 and Hap_1 were the largest haplotypes and they were shared among the three groups (the South China Sea, the Bohai sea, and the Yellow Sea). They are located in the center of the network evolution map, and it is speculated that they are very likely the ancestral haplotypes of the group. Hap_3, which was detected in the *U. meridionalis* samples of the East China Sea, showed limited level of genetic isolation from the other haplotypes; however, it did not reach the degree of subspecies formation (Figure 4). It is speculated that the samples with this haplotype might have experienced a short period of geographical isolation. At the same time, the haplotype network diagram supports the results of the phylogenetic tree. The *U. meridionalis* samples from the East China Sea might be local, but they might also be invasive, either derived from natural conditions or artificially introduced new strains. The *U. meridionalis* samples in the Yellow Sea, the Bohai Sea, and the South China Sea have expanded from haplotype Hap_2 and Hap_1 populations in recent years, and no obvious systematic geographical pattern has been detected among them (Figures 4 and 5).

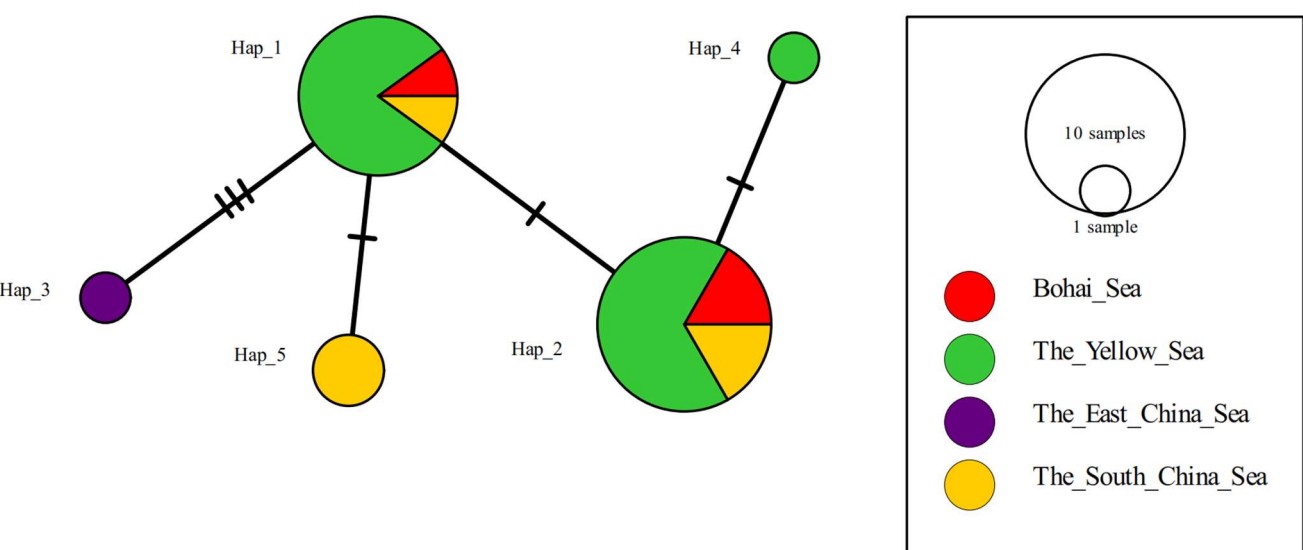

**Figure 5.** Network diagram of the *U. meridionalis* haplotypes derived from the ITS sequences.

## 4. Discussion

Since 1970s, the incidence of green tides has increased steadily [47] and, as the scale of these events continues to expand, prevention has become the focus of attention among researchers. In fact, with the increase in nitrogen supply, the growth rate of *Ulva* spp. also increases [48], and their distribution is expected to expand also due to global warming [49,50]. Therefore, controlling green tide outbreaks is a global challenge. For a long time, the species diversity and the distribution of *Ulva* spp. have attracted considerable attention in terms of the investigation of algal resources, especially as species belonging to this genus can easily cause green tide outbreaks. As more investigations in seaweed farms were conducted worldwide [51], new species or new records were successively reported. For example, *U. meridionalis*, which has a high growth rate with an average daily growth rate of 37% day$^{-1}$ and maximum daily growth rate of more than 112% day$^{-1}$ [34], has been shown to be a new green tide-forming species [34,38,39,52]. At present, there are frequent outbreaks of harmful algal blooms around the world (such as red tides, green tides, and golden tides) [53–55].

As a eurythermal and salt-tolerant species, *U. meridionalis* is widely distributed. At present, it has been found in Asia, Oceania, and North America (Figure 6; Table 1). Up to 20 years ago, this species had been recorded only in Japan. However, in recent years, it has also been reported in the United States, China (the Bohai Sea, the Yellow Sea, the East China Sea, and the South China Sea), Australia, French New Caledonia, and French Polynesia. Furthermore, massive proliferation of this macroalga has been reported in the Bohai Sea, the Yellow Sea, and the South China Sea. At present, *U. meridionalis* blooms as a dominant species in the tropical and subtropical coastal seawaters of China in spring and autumn (Figure 1). In summer, large-scale outbreaks occur even in temperate regions, suggesting that this species has a strong diffusion ability and can gradually spread to higher northern latitudes, which may be due to global warming [50,51,56,57]. The distribution of *U. meridionalis* is increasingly extensive (Table 1); however, the routes through which it is spreading and the diffusion mechanism are unknown, and further studies should investigate these aspects. The discovery of the invasive behavior of *U. meridionalis* in Chinese seas increases the uncertainty of the future trend of green tide outbreaks in various countries.

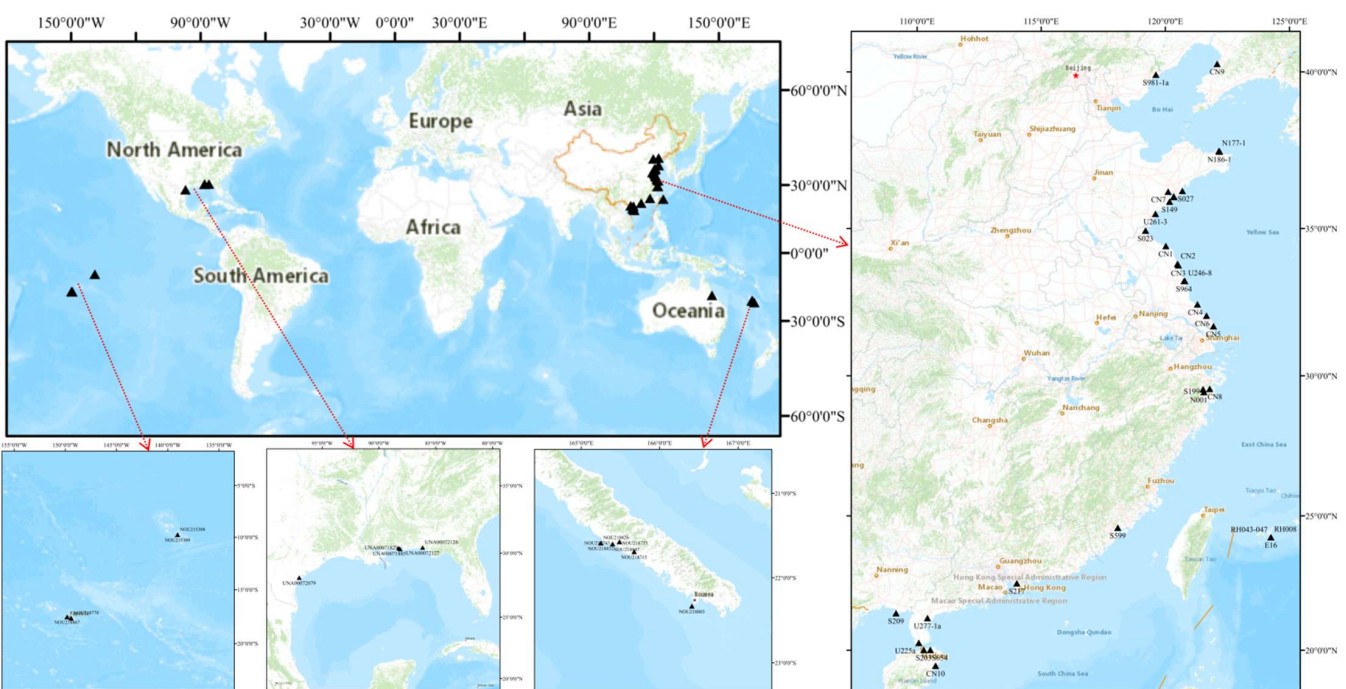

**Figure 6.** Global distribution of *U. meridionalis.* See Table 1 for details on the sampling stations.

*U. meridionalis* has unique and stable morphological characteristics, and it is clearly different from the *Ulva* spp. commonly found in China. At present, there is no obvious evidence that *U. meridionalis* is native to any particular country, therefore the possibility that this macroalga is a new alien invasive species entering China cannot be ruled out. Further molecular studies need to be conducted to trace its source. Previous comparative analyses based on the mitochondrial genome showed that *U. meridionalis* and *U. flexuosa* are genetically close. However, comparative analyses of the chloroplast genome [52,54] revealed that *U. meridionalis* is also closely related to *U. linza* [38]. In the present study, a close genetic relationship was found between *U. meridionalis* and *U. pertusa* based on the ITS sequence. This is inconsistent with the results of previous phylogenetic studies of *U. meridionalis* based on organelle genomes [38,52,54]. In addition, there are relatively few studies on *Ulva* DNA barcode sequences and organelle genomes for the four sea areas in China sampled in the present study (for instance, there is a lack of research on the organelle genome of *U. meridionalis* and other *Ulva* spp. at different sites). At present, the number and type of gene sequences available from the NCBI are not sufficient to effectively determine the genetic relationship of *U. meridionalis* with other *Ulva* species, which remains to be further studied. Genome-wide research [58], plant organelle genome research [40], single nucleotide polymorphism technology, simple sequence repeat technology, DNA barcode sequence research, and other technologies could be adopted at their maximum capacity to clarify the origin of *U. meridionalis* and its outbreak mechanism. These technologies would also allow to explore whether the distribution and outbreaks of this macroalga are related to its own biological characteristics and environmental factors, or they are affected by species variation, intraspecific hybridization, and other associated reasons.

At present, the area with the largest green tide outbreaks in the world is the Southern Yellow Sea. Green tides with *U. prolifera* as the dominant species have occurred in this sea in spring and summer for 15 consecutive years. These events occur on a large scale, and they last long [59]. Whether *U. meridionalis* will bloom under suitable conditions in the Yellow Sea is uncertain. The possibility that this species may also cause large-scale outbreaks similar to *U. prolifera*, harm the structure and function of the local ecosystem, and complicate the green tide dynamics in the Southern Yellow Sea should be the reasons of major concern. In particular, it should be considered that, at present, *U. meridionalis* has

appeared as a floating phenomenon and a large-scale outbreak has been reported in the Bohai Sea (Figure 1A). In addition, this species is also widely distributed in the Yellow Sea, where its biomass is enormous, and large-scale floating has been reported in offshore ponds and intertidal areas as well (Figure 1B).

*U. meridionalis* has a strong ecological adaptability. Studies have shown that at P and N concentrations of 0.125 mg/L and 4.02 mg/L, respectively, its maximum growth rate could reach 79.6% day$^{-1}$ [34]. Compared to the other *Ulva* spp. [47,59–64], *U. meridionalis* shows a very high growth rate. For example, the growth rate of *U. prolifera* in a wide salinity range between 5 and 30 PSU and temperature between 10 and 30°C was 37%-89% day$^{-1}$. In comparison, at a 10–30 salinity range and temperature of 30 °C, the growth rate of *U. meridionalis* could reach 140% day$^{-1}$ [65]. Hiraoka et al. [66] proved that, under the best culture conditions, the growth rate of *U. meridionalis* exceeded 100% day$^{-1}$. Similarly, we found that its daily growth rate was 1.2% day$^{-1}$ under the conditions of 28-30 PSU salinity range and 30 °C temperature, which was also consistent with Hiraoka et al. [66]. Among the multicellular autotrophic plants, *U. meridionalis* shows the fastest growth rate, which makes it a high-quality algal biomass resource [65]. In addition, *U. meridionalis* is characterized by a high photosynthetic carbon sequestration rate, frequent cell proliferation, and rapid formation of algal bodies composed of cell walls in high-temperature environments. The accumulated cell and cell wall components include polysaccharides and rare sugars [65–68]. The prospect of resource utilization is broad. At present, only few studies have been conducted on *U. meridionalis* globally. Just twelve articles (two in Chinese and ten in English) have been published, and the present study mainly reports the physiological and ecological characteristics of this macroalga [14,34,39,40,52,65,69–72]. In general, no taxonomic studies have been conducted on *U. meridionalis*, but fortunately, researchers are gradually beginning to focus on it. For instance, it has been shown that this species has a fast growth rate and a high ammonia nitrogen absorption rate; at P and N concentrations of 3.04 mg/L and 4.26 mg/L, the maximum daily removal rates of $NO_3$-N and $PO_4^{3-}$-P by *U. meridionalis* are 79% and 90%, respectively [34]. It has a broad temperature and salinity resistance and may have several future applications, for example, in the advanced treatment of municipal sewage, in the sulfated polysaccharide extraction industry, and also as a potential pharmaceutical and raw food material [66]. However, its tolerance to toxic heavy metals is unknown. Previous studies have shown that agarophyte *Gracilaria domingensis* (Kutzing) Sonder ex Dickie has strong absorption to Cd and *U. compressa* has strong tolerance to copper [73,74]. However, high concentrations of copper and Cd affect *U. prolifera* gene expression, protein activity, and maximum quantum yield [75]. Therefore, we suggested strengthening the research on the absorption of toxic heavy metals by *U. meridionalis*. Moreover, we also need to further develop new ways to utilize *U. meridionalis* resources by transforming the biomass produced by this green tide-forming species into materials that are beneficial to human life. The use of *U. meridionalis* as a resource will contribute to the reduction of the algal biomass, and, consequently, to the control of potential large-scale outbreaks in the future.

**Author Contributions:** Data curation, M.H., Z.X., Y.T., and J.X.; formal analysis, M.H.; funding acquisition, Z.X.; methodology, J.X. and Y.T.; writing—original draft, M.H.; writing—review and editing, Z.X., Y.T., J.X., S.L., Y.S., J.Z., J.L., S.Z., and J.C. All authors have read and agreed to the published version of the manuscript.

**Funding:** This work is a part of the National Key R&D Program of China (2022YFC3106001, 2022YFC3106004), the Natural Science Foundation of Shanghai (21ZR1427400), and the National Key R & D Program of China (2018YFD0901500).

**Institutional Review Board Statement:** Not applicable.

**Informed Consent Statement:** Not applicable.

**Data Availability Statement:** The datasets generated and/or analyzed in the current study are available from the corresponding author upon reasonable request.

**Conflicts of Interest:** The authors declare no conflict of interest.

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
