# Peer review of "The Morphology, Genetic Diversity, and Distribution of Ulva meridionalis (Ulvaceae, Chlorophyta) in Chinese Seas"

_jmse, doi:10.3390/jmse10121873_

Round 1

Reviewer 1 Report

Dear Editor and Authors,

I have got the opportunity to write a review of the paper titled: "The morphology, genetic diversity, and distribution of Ulva meridionalis in Chinese seas", written by ten authors: Meijuan Hu, Shuang Zhao, Jinlin Liu1, Yichao Tong, Zhangyi Xia, Jing Xia, Shuang Li, Yuqing Sun, Jiaxing Cao, and Jianheng Zhang, from China.

After carefully checking, I can give a positive recommendation for publication in MDPI. Unfortunately, the manuscript needs corrections.

First, a very good paper on this topic was published this year. Xiaoqian Lü, Hao Xu, Sheng Zhao, Fanzhou Kong, Tian Yan, and Peng Jiang. (2022) The green tide in Yingkou, China, in summer 2021 was caused by a subtropical alga—Ulva meridionalis (Ulvophyceae, Chlorophyta). Journal of Oceanology and Limnology 57. Get to know this paper thoroughly.

Unfortunately, the Authors did not follow the journal's high data presentation standards. Below are detailed instructions that can be useful for corrections or resubmissions to other journals. English has to be improved for a more fluent and understandable reading experience. Furthermore, sentences out of context are used to block or interrupt speech at various points throughout the manuscript, sometimes making communication difficult.

My primary concern is that (i) in these modern days, the authors should include more completed data in terms of the profile of other essential chemical elements related to the biodiversity of macroalgae (ie., Ulva) in the Chinese Sea and tolerance by other aquatic organisms (phytoplankton, fish etc.) for hypertrophy and pollution (including toxins, and heavy metals or parameters of water such as salinity) in the coast of China. (ii) The aim(s) of the work is/are unclear; no hypothesis or assumption. The main research objectives must be included in the article (are unclear), especially in the introduction, where it has to be written.

-"Its cells contain starch granules" is it pyrenoids?

- and other cases of "green tides," e.g. in Australia, S America and Europe?

- poor quality of figure 4

- figure 5 (the quality of zoospores and gametes photos are very low)

- more details, please, in the case of the ecology of Ulva (U. prolifera and other important species created: "green tides" in global water) and the influence of "green tide" on the local economy (in the intro)

- Ulva species such as U. pilifera (syn U. flexuosa subsp. pilifera) crate also "green tides" in freshwater ecosystems. Plese, see these papers: 10.1111/j.1529-8817.2011.01048.x, https://doi.org/10.1016/j.ecolind.2020.106951, 10.1111/j.1440-1835.2009.00532.x. These works should be cited.

- Add the author(s) of the taxa description – full name of Ulva species (with citation and only on the first mention) in case o f. U. linza, U. pertusa, U. intestinalis and other species. Please see AlgaBase.

- The ecology of U. meridionalis, on the example of own samples, has been omitted, why?

- Please add the table with physical-chemical data/profile from all China stands (N, P, salinity, temperature, conductivity, TDS and others…)

- Why is there no herbarium documentation (herbarium sheets) of the Ulva meridionalis? Please add the herbarium acronym and methodology. Please see 4 example of methodology used in 10.11646/phytotaxa.345.2.1

I hope that these amendment proposals will be passed in the edition process.

 Best regards,

 Reviewer

Author Response

Response to Reviewer 1 Comments

Dear Editor and Authors,

Point 1: I have got the opportunity to write a review of the paper titled: "The morphology, genetic diversity, and distribution of Ulva meridionalis in Chinese seas", written by ten authors: Meijuan Hu, Shuang Zhao, Jinlin Liu, Yichao Tong, Zhangyi Xia, Jing Xia, Shuang Li, Yuqing Sun, Jiaxing Cao, and Jianheng Zhang, from China.

After carefully checking, I can give a positive recommendation for publication in MDPI. Unfortunately, the manuscript needs corrections.

First, a very good paper on this topic was published this year. Xiaoqian Lü, Hao Xu, Sheng Zhao, Fanzhou Kong, Tian Yan, and Peng Jiang. (2022) The green tide in Yingkou, China, in summer 2021 was caused by a subtropical alga—Ulva meridionalis (Ulvophyceae, Chlorophyta). Journal of Oceanology and Limnology 57. Get to know this paper thoroughly.

Unfortunately, the Authors did not follow the journal's high data presentation standards. Below are detailed instructions that can be useful for corrections or resubmissions to other journals. English has to be improved for a more fluent and understandable reading experience. Furthermore, sentences out of context are used to block or interrupt speech at various points throughout the manuscript, sometimes making communication difficult.

I hope that these amendment proposals will be passed in the edition process.

Best regards,

Reviewer

Response 1: Thank you very much for your suggestion. We have read the article you recommended. This article reported the green tide caused by Ulva meridionalis in Yingkou, Dalian in 2021. This event is also the source of inspiration for our article. We also think this article is very interesting and wonderful, we have learned a lot from this article. Therefore, we have cited it in the discussion and revised the question you raised, and the added supplement section has been fully supplemented. We believe that your professional suggestions will make our article further improved. Thank you sincerely. As for the language question you raised, we have sent the manuscript to International Science Editing for language modification before submission. We have also modified some phrases that are not clear. Thank you again for your wonderful comments.

Point 2: My primary concern is that (i) in these modern days, the authors should include more completed data in terms of the profile of other essential chemical elements related to the biodiversity of macroalgae (ie., Ulva) in the Chinese Sea and tolerance by other aquatic organisms (phytoplankton, fish etc.) for hypertrophy and pollution (including toxins, and heavy metals or parameters of water such as salinity) in the coast of China. (ii) The aim(s) of the work is/are unclear; no hypothesis or assumption. The main research objectives must be included in the article (are unclear), especially in the introduction, where it has to be written.

Response 2: This study introduces the basic biological characteristics of U. meridionalis, the species relationship of U. meridionalis in the four sea areas of China, and the current distribution of U. meridionalis in the world, so there is no research on macroalgae biodiversity in relation to chemical elements and tolerance of aquatic organisms to coastal pollution in China. However, we added in the introduction about the toxic release of harmful algae caused by environmental deterioration in Chinese sea, and added in the discussion about the tolerance of aquatic plants to heavy metals, hoping to make the article more complete.

Point 3: -"Its cells contain starch granules" is it pyrenoids?

Response 3: The starch granules I wrote in the article is indeed pyrenoids. It was my translation error at the beginning. Thank you for your question and I have made the correction.

Point 4: - and other cases of "green tides," e.g. in Australia, S America and Europe?

Response 4: We add examples of green tide outbreaks in other countries at Lines 57-59.

Point 5: - poor quality of figure 4

Response 5: Because the pdf exported by MDPI under review is a low-quality version, actually we uploaded the HD version, and we have adjusted the drawing. MDPI Press will publish the paper with high-definition pictures when it is published. Thank you very much for your comments.

Point 6: - figure 5 (the quality of zoospores and gametes photos are very low)

Response 6: The photos were produced by our laboratory microscope in high definition, and the photos we uploaded are also in high definition.

Point 7: - more details, please, in the case of the ecology of Ulva (U. prolifera and other important species created: "green tides" in global water) and the influence of "green tide" on the local economy (in the introduction)

Response 7: Line 41-48, we added the impact of green tide outbreak on marine organism, ecosystem and economy to improve the harm of green tide. Thank you for your comments.

Point 8: - Ulva species such as U. pilifera (syn U. flexuosa subsp. pilifera) crate also "green tides" in freshwater ecosystems. Plese, see these papers: 10.1111/j.1529-8817.2011.01048.x, https://doi.org/10.1016/j.ecolind.2020.106951, 10.1111/j.1440-1835.2009.00532.x. These works should be cited.

Response 8: We have cited all the paper. Thank you.

Point 9: - Add the author(s) of the taxa description – full name of Ulva species (with citation and only on the first mention) in case of. U. linza, U. pertusa, U. intestinalis and other species. Please see AlgaBase.

Response 9: We've changed them. That’s great suggestion.

Point 10: - The ecology of U. meridionalis, on the example of own samples, has been omitted, why?

Response 10: Your consideration is very comprehensive, and we have added our study on the growth rate of U. meridionalis at Line 334-340.

Point 11: - Please add the table with physical-chemical data/profile from all China stands (N, P, salinity, temperature, conductivity, TDS and others…)

Response 11: Dear reviewer, your suggestions are worth referring to, because the outbreak of U. meridionalis formed in Yingkou, Dalian, China in 2021, it began to attract extensive attention of researchers.In this paper, we study U. meridionalis on the basic biology, and use molecular identification and phylogenetic methods to study the species relationship of U. meridionalis in four sea areas of China and summarize the distribution of U. meridionalis in the world. At the same time, this paper reports the outbreak of small scale green tide or algal mat in the Bohai Sea, the Yellow Sea and the South China Sea caused by U. meridionalis, in order to arouse the academic attention to the possible outbreak of large-scale green tide of U. meridionalis in China. Therefore, this article does not involve the environmental factors of China sea on U. meridionalis. Of course, we are also very interested in the direction you proposed. For the influence of environmental factors on U. meridionalis, such as the influence of temperature and salinity changes on the growth rate and physiological characteristics of U. meridionalis, we will discuss in the next article. Thank you very much for your wonderful comments, we learned a lot from them.

Point 12: - Why is there no herbarium documentation (herbarium sheets) of the Ulva meridionalis? Please add the herbarium acronym and methodology. Please see 4 example of methodology used in 10.11646/phytotaxa.345.2.1

Response 12: We have added the methodology of herbarium acronym at lines 131-135 and herbarium acronym at lines 187-188. Thank you for your excellent advice.

Reviewer 2 Report

The paper is quite interesting and deserves to be published. However, some modifications reported in the text should be made.

Author Response

Response to Reviewer 2 Comments

Point 1: Line 3, Title Part, add ‘(Ulvaceae, Chlorophyta)’; Line 17, ‘sp. nov., a new Ulva species belonging to the Ulvaceae family,’. The species was described in 2011 and therefore ‘sp. nov.’ should be omitted.

Response 1: Thank you for your comments, we have revised them.

Point 2: Line 28, Since the species was described in 2011, how is possible that its distribution area widened after 2007?

Response 2: Ulva meridionalis was first reported in 2011, in Australia in 2016, in China in 2018 and 2021, in the United States in 2020, and in Oceania in 2022, so we changed the year to 2011. Thank you very much.

Point 3: Line 50, ‘more than 70’. Now about 100. Guiry, M.D. & Guiry, G.M. 2022. AlgaeBase. World-wide electronic publication, National University of Ireland, Galway. https://www.algaebase.org. (to be added in References).

Response 3: Your suggestions are very wonderful, we have made changes.

Point 4: Line 53, Change to ‘Ulva prolifera O.F. Müller,’, ‘Ulva compressa Linnaeus,’, ‘Ulva intestinalis Linnaeus,’, ‘Ulva flexuosa Wulfen,’ ‘Ulva australis Areschoug [16-20 (as U. pertusa Kjellman].’.

Response 4: Thank you for your very comprehensive suggestions, we have corrected them.

Point 5: Line 56, Delete ‘, sp. nov. (U. meridionalis)’. Delete ‘is a green alga belonging to the order Ulvales, family Ulvaceae, and Ulva genus [21]. It’.

Response 5: we have deleted them.

Point 6: Line 62, ‘a large amount’ change to ‘a large amount of’; Line 76, ‘Xie et al., 2020’ need to change cite style; Line 110, ‘branching’ change to ‘thallus’; Line 111, ‘in the base’ change to ‘at the basal’; Line 113, ‘sporangium, tetraflagellate meiospore’ change to ‘release of tetraflagellate meiospore’; Line 114, Delete ‘produced by promoting gamete release’; Line 133, ‘U. pertusa’ need to in italics. Line 134, Change to ‘U. erecta (Lyngbye) Fries’. Line 135, Change to ‘U. tepida Y. Masakiyo et S. Shimada’; Line 151, ‘branch structure’ change to ‘axis’, ‘especially’ change to ‘that’

Response 6: Thank you for your very careful suggestions, we have changed all of them.

Point 7: Line 134, You must specify the author of U. fasciata. In fact, there were described two species with the same name: U. fasciata Delile (synonym of U. lactuca Linnaeus) and U. fasciata S.F. Gray nom. illeg.(synonym of U. linza Linnaeus).

Response 7: We have determined its species name.

Point 8: Table 1 ‘Attached to the intertidal zone’ ‘Attached to the pond’ ‘Attached to the intertidal mudflat’ ‘Attached to the estuary’ change to ‘Attached in the intertidal zone’ ‘Attached in the pond’ ‘Attached in the intertidal mudflat’ ‘Attached in the estuary’. ‘Attached to the seawall or pier’ change to ‘Attached to a seawall or pier’.

Response 8: Thanks. We have corrected them all.

Point 9: Line 152, If the structure is monolayered it consists of only one layer of cells. Therefore, it is not clear why you say that there is a double-layer cell structure.

Response 9: Thanks for your question, I've replaced the original statement with “With the growth of the algae, the middle of the tubular algae flattens and the upper and lower layers of cells stick together. In transverse section of the middle of the algae is double-layer cell structure, and the edge is still monolayered tubular structure.”

Point 10: Line 153, Such a phrase is not clear. What do you mean for “transverse branch”? Maybe basal proliferations? Please re-write.

Response 10: Your question is very meaningful. The branches at the base are nearly parallel and perpendicular to the axis, these branches are called transverse branche. It doesn't mean basal proliferations.

Point 11: Line 158, Delete ‘also’; Line 159, Delete ‘moving’; Line 161, ‘base’ change to ‘basal’ and delete ‘algal’.

Response 11: We have made changes. Thank you.

Point 12: Line 161, How many pyrenoids did you observe?

Response 12: That’s a key question. We have observed 122 chloroplasts.

Point 13: Line 164, ‘forms spores’ change to ‘releases reproductive cells’; Line 165-168, change to ‘and new sporophytes or gametophytes can form. The biflagellate gametes [which form in the gametangium (Fig. 2H)] are oval, have a reddish eyespot (Fig. 2J-K) and show positive phototaxis; tetraflagellate meiospores are ovate and have a reddish eyespot (Fig. 2I), and show a negative phototaxis.’.

Response 13: Thank you for your essential suggestions, we changed them.

Point 14: Line 170, (Fig. 2A) It isn’n clear what figure shows; ‘macroalga’ change to ‘thallus’. Line 172, change to ‘main branch’‘(D) surface view of cells in the basal portion’ ‘more’ change to ‘many’ ‘(F) surface view of cells in the middle region’ ‘(G) surface view of cells in the upper region’; Line 182, change to ‘although’; Line 183, Change to ‘barcode’. Line 251-252, Delete ‘The discovery of U. meridionalis not …’

Response 14: Your suggestions are very fantastic, we have changed.

Point 15: Line 267, ‘The discovery of invasive behavior of U. meridionalis in Chinese seas increases the uncertainty of the future trend of green tide outbreaks in various countries.’ need to add; Line 275, Delete ‘the relatives of’; Line 277, Change to ‘U. meridionalis is also closely’; Line 282, ‘for instance’; Line 309, indicate the units of measurement of salinity, PSU?; Line 322, ‘systematic’ change to ‘taxonomic or physiological’; Line 323, ‘, but fortunately,’

Response 15: We have made the modification according to your suggestion.

Point 16: Reference 56 and 61 not quoted in the text.

Response 16: Thank you very much for your sweet, we have made modifications.

Reviewer 3 Report

Dear colleagues,

The peer-reviewed article jmse-2051439 is the result of a successful combination of classic morphological-taxonomic and modern molecular approaches to the study of the diversity and ecological-geographic features of species of the genus Ulva. The polyphasic criterion made it possible to identify a new species of this genus, to show its phylogenetic relationship with sister species, and to reveal taxonomic specificity. Based on the volume of the material and the results obtained, the reviewed article deserves to be published on the pages of your journal, but with minor revisions:

l. 309-311 - indicate the units of measurement of salinity and growth rate;

l. 324-325 - it is desirable to indicate (for comparison with the results of the author's article) previously known indicators of growth rate and nitrogen absorption rate as unique characteristics of the species

Author Response

Response to Reviewer 2 Comments

Dear colleagues,

The peer-reviewed article jmse-2051439 is the result of a successful combination of classic morphological-taxonomic and modern molecular approaches to the study of the diversity and ecological-geographic features of species of the genus Ulva. The polyphasic criterion made it possible to identify a new species of this genus, to show its phylogenetic relationship with sister species, and to reveal taxonomic specificity. Based on the volume of the material and the results obtained, the reviewed article deserves to be published on the pages of your journal, but with minor revisions:

Point 1: l. 309-311 - indicate the units of measurement of salinity and growth rate;

Response 1: Thank you very much for your suggestion. We have added the salinity unit PSU and changed the unit of growth rate to the percent sign.

Point 2: l. 324-325 - it is desirable to indicate (for comparison with the results of the author's article) previously known indicators of growth rate and nitrogen absorption rate as unique characteristics of the species

Response 2: Your suggestion is particularly useful. We have added the maximum daily N and P removal rate of Ulva meridionalis. Your suggestions will make our article more exciting. Thank you very much for your comments